# Untangling Cosmic Magnetic Fields: Faraday Tomography at Metre Wavelengths with LOFAR

**Shane P. O'Sullivan** [1,*], **Marcus Brüggen** [1], **Cameron L. Van Eck** [2], **Martin J. Hardcastle** [3], **Marijke Haverkorn** [4], **Timothy W. Shimwell** [5], **Cyril Tasse** [6], **Valentina Vacca** [7], **Cathy Horellou** [8] **and George Heald** [9]

1   Hamburger Sternwarte, Universität Hamburg, Gojenbergsweg 112, D-21029 Hamburg, Germany; mbrueggen@hs.uni-hamburg.de
2   Dunlap Institute for Astronomy and Astrophysics, University of Toronto, 50 St. George Street, Toronto, ON M5S 3H4, Canada; cameron.van.eck@dunlap.utoronto.ca
3   Centre for Astrophysics Research, School of Physics, Astronomy and Mathematics, University of Hertfordshire, College Lane, Hatfield AL10 9AB, UK; m.j.hardcastle@herts.ac.uk
4   Department of Astrophysics/IMAPP, Radboud University Nijmegen, P.O. Box 9010, 6500 GL Nijmegen, The Netherlands; m.haverkorn@astro.ru.nl
5   ASTRON, The Netherlands Institute for Radio Astronomy, Postbus 2, 7990 AA Dwingeloo, The Netherlands; shimwell@astron.nl
6   GEPI & USN, Observatoire de Paris, Université PSL, CNRS, 5 Place Jules Janssen, 92190 Meudon, France; cyril.tasse@obspm.fr
7   INAF—Osservatorio Astronomico di Cagliari, Via della Scienza 5, 09047 Selargius (CA), Italy; vvacca@oa-cagliari.inaf.it
8   Department of Space, Earth and Environment, Chalmers University of Technology, Onsala Space Observatory, 43992 Onsala, Sweden; cathy.horellou@chalmers.se
9   CSIRO Astronomy and Space Science, P.O. Box 1130, Bentley 6102, Australia; George.Heald@csiro.au
*   Correspondence: shane@hs.uni-hamburg.de

**Abstract:** The technique of Faraday tomography is a key tool for the study of magnetised plasmas in the new era of broadband radio-polarisation observations. In particular, observations at metre wavelengths provide significantly better Faraday depth accuracies compared to traditional centimetre-wavelength observations. However, the effect of Faraday depolarisation makes the polarised signal very challenging to detect at metre wavelengths (MHz frequencies). In this work, Faraday tomography is used to characterise the Faraday rotation properties of polarised sources found in data from the LOFAR Two-Metre Sky Survey (LoTSS). Of the 76 extragalactic polarised sources analysed here, we find that all host a radio-loud AGN (Active Galactic Nucleus). The majority of the sources (∼64%) are large FRII radio galaxies with a median projected linear size of 710 kpc and median radio luminosity at 144 MHz of $4 \times 10^{26}$ W Hz$^{-1}$ (with ∼13% of all sources having a linear size >1 Mpc). In several cases, both hotspots are detected in polarisation at an angular resolution of ∼20″. One such case allowed a study of intergalactic magnetic fields on scales of 3.4 Mpc. Other detected source types include an FRI radio galaxy and at least eight blazars. Most sources display simple Faraday spectra, but we highlight one blazar that displays a complex Faraday spectrum, with two close peaks in the Faraday dispersion function.

**Keywords:** magnetic fields; Faraday tomography; large-scale structure; AGN; Milky Way

## 1. Introduction

Advances in radio-receiver technology that enable wide-spanning observations, continuous frequency ranges, at a high spectral resolution, have opened a new parameter space for studies

of cosmic magnetism. In particular, they bring the technique of Faraday tomography to the fore, which allows detailed studies of the distribution of linearly polarised radiation as a function of Faraday depth, $F(\phi)$, which probes the physical properties of magnetised plasma along the line of sight. In this formalism, complex linearly polarised intensity $P(\lambda^2)$ is expressed as:

$$P(\lambda^2) = \int_{-\infty}^{\infty} F(\phi)\, e^{2i\phi\lambda^2}\, d\phi, \tag{1}$$

where $F(\phi)$ is commonly known as the Faraday dispersion function (FDF), and Faraday depth $\phi = 0.81 \int_{L}^{0} n_e B_{||} dl \,\mathrm{rad\,m^{-2}}$ encodes the amount of Faraday rotation caused by a magnetoionic region of electron number density ($n_e$, cm$^{-3}$), line-of-sight magnetic field strength ($B_{||}$, $\mu$G), and path length ($l$, pc).

Since the accuracy with which one can reconstruct $F(\phi)$ depends on the total wavelength-squared ($\lambda^2$) coverage, broadband polarisation observations at metre wavelengths [1–4] can provide a typical improvement close to two orders of magnitude in Faraday depth accuracy compared to observations at centimetre wavelengths [5–8]. However, the distribution of Faraday depths within the synthesised beam of the telescope can cause Faraday depolarisation, which makes the polarised signal fainter and more difficult to detect at long wavelengths [9].

In the context of the linearly polarised synchrotron emission from extragalactic radio-loud AGN, there are many possible contributions to the net observed Faraday rotation measure (RM) and Faraday depolarisation along the line of sight: from magnetised plasma internal to the radio-emitting source (e.g., Reference [10]), from the turbulent magnetic fields associated with the hot ionised gas of the ambient group or cluster environment (e.g., Reference [11]), magnetoionic gas potentially associated with the filamentary large-scale structure of the universe outside of clusters of galaxies (e.g., Reference [12]), the magnetised disks and halos of intervening galaxies (e.g., Reference [13]), the magnetised interstellar medium of the Milky Way (e.g., Reference [14]), and the time-variable Faraday rotation contribution from the Earth's ionosphere (e.g., Reference [15]). The technique of Faraday tomography at radio wavebands, in addition to complementary observations at other wavebands, is a powerful tool to help isolate the contributions from all these regions of magnetoionic material.

The Faraday tomography results presented in this paper are based on data from the Low-Frequency Array (LOFAR [16]), which is a radio interferometer capable of observing from 10 to 90 MHz with Low-Band Antennas (LBA) and from 110 to 250 MHz with High-Band Antennas (HBA). The array is composed of 'core stations' that provide many short baselines, up to $\sim$2 km, with more sparsely distributed 'remote stations' extending out to $\sim$100 km from the core. There are also several international stations spread throughout Europe that can provide baselines of $\sim$1000 km. In particular, we make use of data from the ongoing LOFAR Two-Metre Sky Survey (LoTSS [17]), coupled with a preliminary catalog of linearly polarised sources [18]. The LoTSS survey aims to observe the whole northern sky above $0°$ declination using the LOFAR HBA from 120 to 168 MHz.

LoTSS observations are currently ongoing ($\sim$20% complete), but there is an initial data release (LoTSS-DR1) of images and catalogs for 325,694 radio sources, detected 5 times above the noise level, covering 424 square degrees in the HETDEX Spring Field (RA: $10^\mathrm{h}45^\mathrm{m}$ to $15^\mathrm{h}30^\mathrm{m}$, Dec: $45°$ to $57°$) [19]. The median sensitivity achieved in images of this region is $\sim$70 $\mu$Jy beam$^{-1}$, at an angular resolution of $6''$, and diffuse extended radio emission can be recovered with high fidelity. This provides a radio-source surface density $\sim$10 times higher than the best existing area-wide radio surveys and is comparable, for typical radio sources, to that expected from the planned ASKAP-EMU [20] and APERTIF [21] surveys. The LoTSS-DR1 release also comes with host-galaxy identifications for 72% of the radio sources, and provides spectroscopic and photometric redshifts for 70% of these sources [22,23].

The polarisation catalog of this region found 92 polarised radio components (including 1 pulsar) after imaging at an angular resolution of $4.3'$ [18]. The focus of this paper is on the further investigation of the physical properties of extragalactic polarised sources, making use of the calibrated data, images

(20″ and 6″ angular resolution), and value-added data products (radio-source flux densities, host identifications, redshifts, and source sizes) provided by the LoTSS-DR1 team.

## 2. Methods

Radio-visibility data were calibrated using the PREFACTOR pipeline https://github.com/lofar-astron/prefactor [15,17], which includes the time-dependent ionospheric Faraday rotation correction using RMEXTRACT https://github.com/lofar-astron/RMextract [24]. The estimated residual ionospheric RM correction errors are ∼0.1 to 0.3 rad m$^{-2}$ [18,25]. To create Stokes $Q$ and $U$ cubes for each of the known polarised sources, the calibrated data were phase-shifted to the source position and averaged in time to reduce the data size using NDPPP [26] https://support.astron.nl/LOFARImagingCookbook/. Imaging software WSCLEAN [27] https://sourceforge.net/projects/wsclean was used to create channel images at 97.6 kHz resolution, with a minimum uv-range of 150$\lambda$ and a maximum UV-range of 18k$\lambda$, producing channel images with an angular resolution of ∼20″. RM synthesis and RMCLEAN [28,29] were applied using PYRMSYNTH https://github.com/mrbell/pyrmsynth to create RM cubes with a range of ±150 rad m$^{-2}$, sampled at 0.15 rad m$^{-2}$. The frequency range of 120 to 168 MHz, with a channel resolution of 97.6 kHz, provides a theoretical RM resolution of ∼1.1 rad m$^{-2}$, with a maximum scale of ∼1 rad m$^{-2}$, and a maximum detectable |RM| of ∼450 rad m$^{-2}$. See References [18,30,31] for full details of the polarisation and Faraday rotation imaging and analysis methods, and Reference [19], and references therein, for the data-calibration and total intensity-imaging procedures.

## 3. Results

### 3.1. Polarisation and Faraday Rotation Properties

Of the 91 extragalactic polarised sources found by Reference [18] in a region of 570 deg$^2$, 76 reside within the LoTSS-DR1 release area of 424 deg$^2$ [19]. It is the properties of these 76 sources that we present in this paper. This is only ∼0.4% of the 19,233 radio sources with total flux densities greater than 10 mJy in the LoTSS-DR1 area. This emphasises the scarcity of polarised sources at 144 MHz, with a polarised source sky density of ∼0.18 per square degree[1]. The faintest polarised source was detected at ∼0.8 mJy beam$^{-1}$, and the brightest at ∼98 mJy beam$^{-1}$, with the majority of sources detected around 3.5 mJy beam$^{-1}$ (Figure 1, left panel).

A comparison with the RM of the same sources at 1.4 GHz [32], derived from the NRAO VLA Sky Survey (NVSS [33]), found that the LOFAR RM distribution was significantly narrower, likely due to the smaller errors, but also because sources near Faraday depths of 0 rad m$^{-2}$ are missing due to contamination from uncorrected instrumental polarisation, making the LOFAR RM catalog incomplete [18]. The scatter in the difference between RM values derived at 1.4 GHz and 144 MHz was also found to be significantly larger than expected from the derived errors, suggesting that regions of large RM variance seen at 1.4 GHz were completely depolarised at 144 MHz [18].

The integrated degree of polarisation at 144 MHz was also found to be lower than at 1.4 GHz in all cases, as expected due to the effect of Faraday depolarisation. However, even though the median degree of polarisation is ∼8 times lower at 144 MHz than at 1.4 GHz [18], there were still 33 polarised sources found at 144 MHz that were not in the NVSS RM catalog [32]. This is due to a combination of small amounts of Faraday depolarisation for some sources, coupled with the LoTSS survey being much deeper than the NVSS. The noise level in Stokes $Q$ and $U$ in the LoTSS data

---

[1]   Note that sources with an RM near zero (±2.5 rad m$^{-2}$) have been excluded due to contamination from instrumental polarisation in this range, and sources with an RM magnitude greater than ∼450 rad m$^{-2}$ are strongly affected by bandwidth depolarisation due to the channel spacing of 97.6 kHz. Furthermore, higher polarised source densities are expected at higher angular resolution, as demonstrated in a pointed observation of one of the fields by [31], which found a source density of ∼0.3 per deg$^2$ at ∼18″.

at 4.3′ was ∼0.15 mJy beam$^{-1}$ [18], which was ∼10 times more sensitive than the NVSS for steep spectrum-polarised radio sources (assuming a spectral index of −0.7). In addition, the NVSS RM catalog uses an $8\sigma_{QU}$ cutoff compared to the ∼$5\sigma_{QU}$ threshold used at 144 MHz.

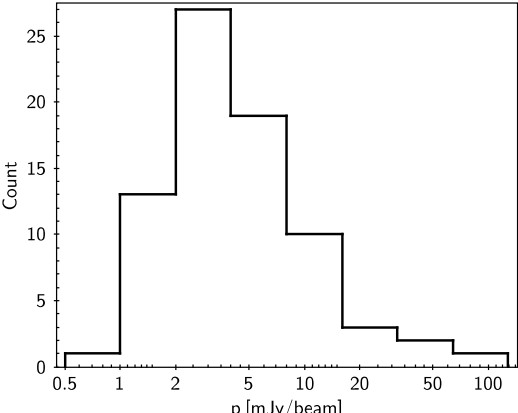 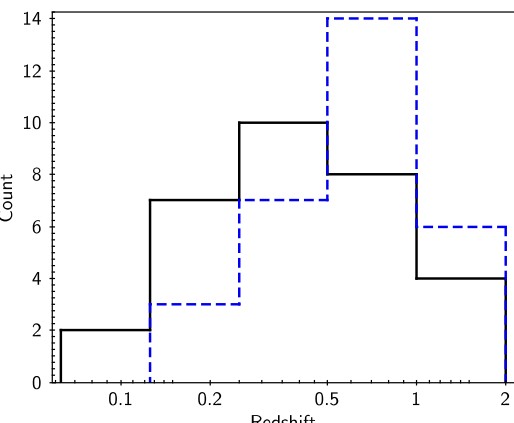

**Figure 1.** (**Left panel**) histogram of the peak polarised intensity of the 76 polarised sources in LOFAR Two-Metre Sky Survey (LoTSS)-DR1; (**right panel**) histograms of spectroscopic (solid line) and photometric (dashed blue line) redshifts of the polarised sources.

### 3.2. Radio Luminosity, Morphology, and Projected Linear Size

Based on the photometric [23] and spectroscopic [22] redshift catalogs of LoTSS-DR1, there were redshifts for ∼80% of the 76 polarised sources (31 spectroscopic redshifts and 30 photometric redshifts), while eight of the sources without a redshift also lacked an optical identification of the host galaxy. All identified sources were consistent with being radio-loud AGN. The spectroscopic and photometric redshift distributions of the polarised sources are shown in Figure 1, right panel. They range from z ∼ 0.1 to z ∼ 1.5, with a median redshift of 0.5. This is similar to the redshift distribution of all radio-loud AGN in the LoTSS-DR1 area [34]. The radio-luminosity distribution at 144 MHz ($L_{144\,MHz}$) confirms the nature of these sources as powerful radio-loud AGN. This ranged from $3.6 \times 10^{24}$ W Hz$^{-1}$ to $1 \times 10^{28}$ W Hz$^{-1}$ with a median luminosity of $4 \times 10^{26}$ W Hz$^{-1}$ (Figure 2, left panel).

The majority of the polarised sources are resolved in the 6″ total intensity images (62/76), with a median angular size for all polarised sources of 73″. From inspection of the total intensity images, the majority of the polarised sources can be identified as FRII morphology radio galaxies (49/76), with the same luminosity range and median as above. One FRI radio galaxy and at least eight blazars were identified, while the remaining compact or morphologically ambiguous sources require higher angular-resolution observations to determine their source type. The projected linear size can be calculated for 49 of the resolved sources. Linear-size distribution is shown in Figure 2 (right), with upper limits included for the unresolved sources. The median linear size of all sources is 415 kpc with a range from <50 kpc up to 3.4 Mpc. As a large fraction of sources (10/76) have a linear size >1 Mpc, polarisation observations at low frequencies can be useful in selecting for 'giant' radio galaxies. In addition to the main population of FRII sources (with the median linear size for the 39 of those FRIIs with redshifts being 720 kpc), there is also a smaller population of compact sources, the majority of which can be identified as blazars [35].

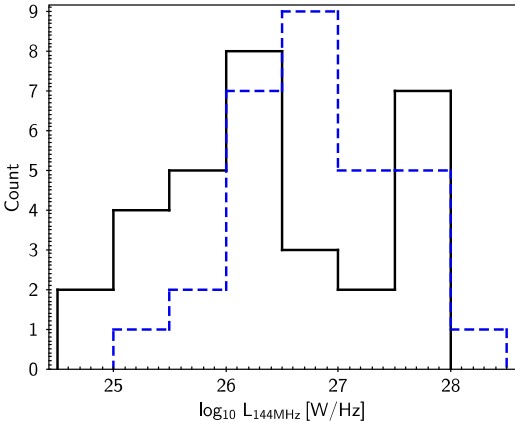 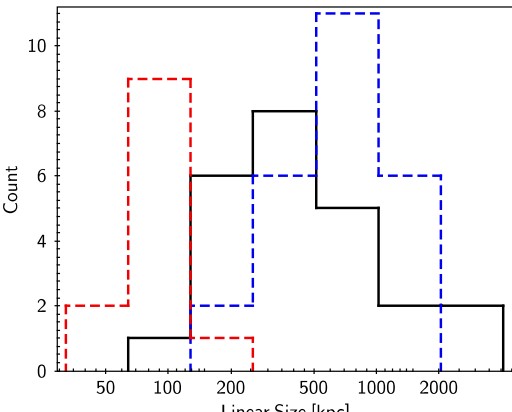

**Figure 2.** (**Left panel**) histograms of radio luminosity at 144 MHz ($L_{144\,MHz}$) for the polarised sources with spectroscopic (solid line) and photometric (dashed blue line) redshifts. Radio luminosities are consistent with all sources being radio-loud AGN; (**right panel**) histograms of the projected linear size of resolved sources with spectroscopic (solid line) and photometric (blue dashed line) redshifts. The red dashed line shows the histogram of upper limits on the projected linear size of unresolved sources with redshift measurements.

### 3.3. In-Depth Study of the Faraday Rotation from the Largest FRII Radio Galaxy in the Sample

The source with the largest projected linear size of 3.4 Mpc (and angular size of 11'), ILT J123459.82+531851.0, at a redshift of z ∼ 0.34 has been investigated in detail (see Reference [30] for a full description). Here, we present a brief summary of the results from that study. The key result was based on the polarisation and RM distribution of the lobes at an angular resolution of 20″ (Figure 3). A mean RM difference of ∼2.5 rad m$^{-2}$ was found between the regions of polarised emission in the opposite lobes of this FRII radio galaxy. This RM difference was investigated in the context of the potential contribution of intergalactic magnetic fields (IGMF) in foreground large-scale-structure (LSS) filaments.

Due to the large linear size of the source, the polarised emission from the northwest and southeast lobes probes significantly different cosmic lines of sight, such that an excess of up to three LSS filaments were estimated to be probed by the polarised emission from the northwest lobe. The LSS filaments were identified from a catalog constructed using spectroscopic observations of galaxies in the Sloan Digital Sky Survey (SDSS) [36,37]. Associating the entire RM difference to these LSS filaments implies a gas density-weighted magnetic field strength of ∼0.3 µG. However, from comparisons with cosmological MHD simulations [38] of the expected RM signal from LSS filaments, it was found that an RM difference as large as 2.5 rad m$^{-2}$, on 3.4 Mpc scales, had a low probability (∼5%) of occurring, for IGMF strengths of tens of nG. The magnetic field in the simulation was amplified to between 10 and 50 nG from an initial magnetic field of 1 nG seeded at early cosmological epochs, which is close to the upper limit for such fields provided by the Planck satellite [39].

Furthermore, variations in the Faraday rotation of the Milky Way on scales of 11' likely contribute significantly to the observed RM difference between the lobes. The best available reconstructions of the Galactic RM [40] have a resolution of only ∼1 degree, although RM structure function analyses suggest RM variations of several rad m$^{-2}$ are possible on arcminute scales [14,41,42]. This highlights the need for denser RM grids that are expected from VLASS [43] and ASKAP-POSSUM [44] in the near future, to provide a better understanding of the Galactic RM variations on scales less than 1 degree [45].

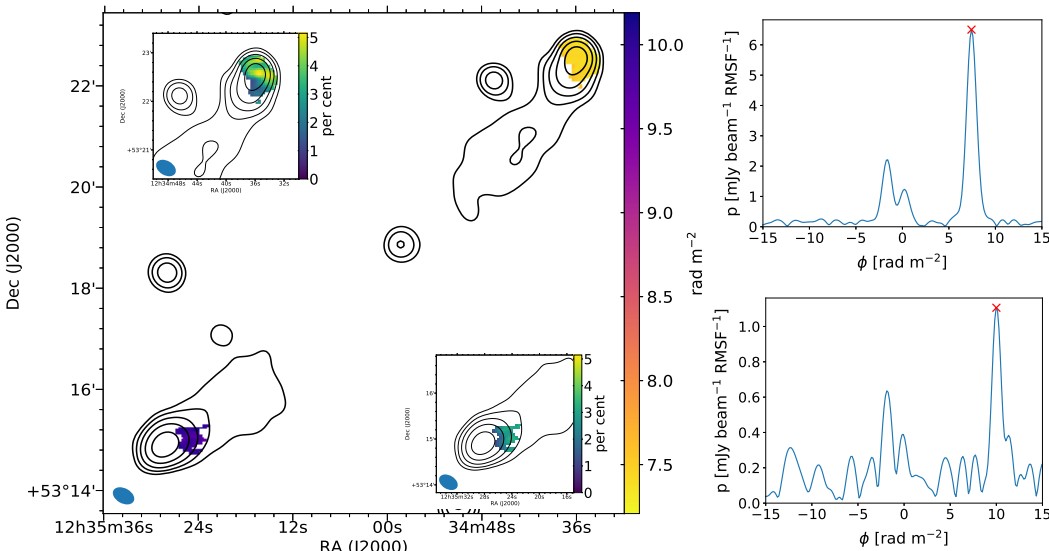

**Figure 3.** Faraday rotation and degree of polarisation image for ILT J123459.82+531851.0 (described in Section 3.3), adapted from Reference [30]. (**Main image**) Total intensity contours at 25″ with the colour scale showing the Faraday rotation measure (RM) of regions detected in polarisation; (**insets**) zoom-in on the NW and SE lobes, with the degree of polarisation shown with the colour scale; (**top right**) Faraday dispersion function (FDF) for the peak polarised intensity in the NW lobe. The red cross marks the peak of the source-related polarised emission. The other peaks are either noise peaks or related to instrumental polarisation; (**bottom right**) FDF for the brightest polarised intensity pixel of the SE lobe.

### 3.4. High-Resolution Faraday Rotation Imaging with LOFAR

In the preliminary catalog, polarised sources from the HETDEX region of the LoTSS survey data were imaged at a relatively low angular resolution of 4.3′ [18]. One of the reasons for imaging at low angular resolution was due to the challenging computational requirements for processing and storing polarisation data products at a high angular resolution across the entire LoTSS area. To circumvent this limitation, smaller datasets have been made for the polarised sources (by phase-shifting the UV data to the source co-ordinates and averaging in time), which were then imaged at the higher angular resolution of 20″, following the procedures described in Reference [30,31]. Figure 4 shows examples of the RM distribution in these high-resolution images, where the grey total intensity contours were at the same resolution as the RM image (20″), while the black total intensity contours were at 6″.

Figure 4 (top left) shows a large angular (∼500″) and linear size (∼1.9 Mpc) FRII radio galaxy ($L_{144\,\mathrm{MHz}} \sim 7 \times 10^{25}$ W Hz$^{-1}$), with the polarised hotspots displaying a difference in RM of ∼2.1 rad m$^{-2}$. This demonstrates the impressive capability of the LoTSS survey to image both compact and extended regions of large radio galaxies. Another advantage of imaging at higher angular resolution is that only the southwest hotspot was previously detected in polarisation [18]. Figure 4 (top right) shows another example of an FRII radio galaxy ($L_{144\,\mathrm{MHz}} \sim 5 \times 10^{26}$ W Hz$^{-1}$), which is the most common morphology of the polarised sources currently found at 144 MHz [18], with 64% of polarised sources being identified as FRIIs. Generally, polarised hotspots have simple Faraday spectra, with only a single peak in the FDF (e.g., Figure 5).

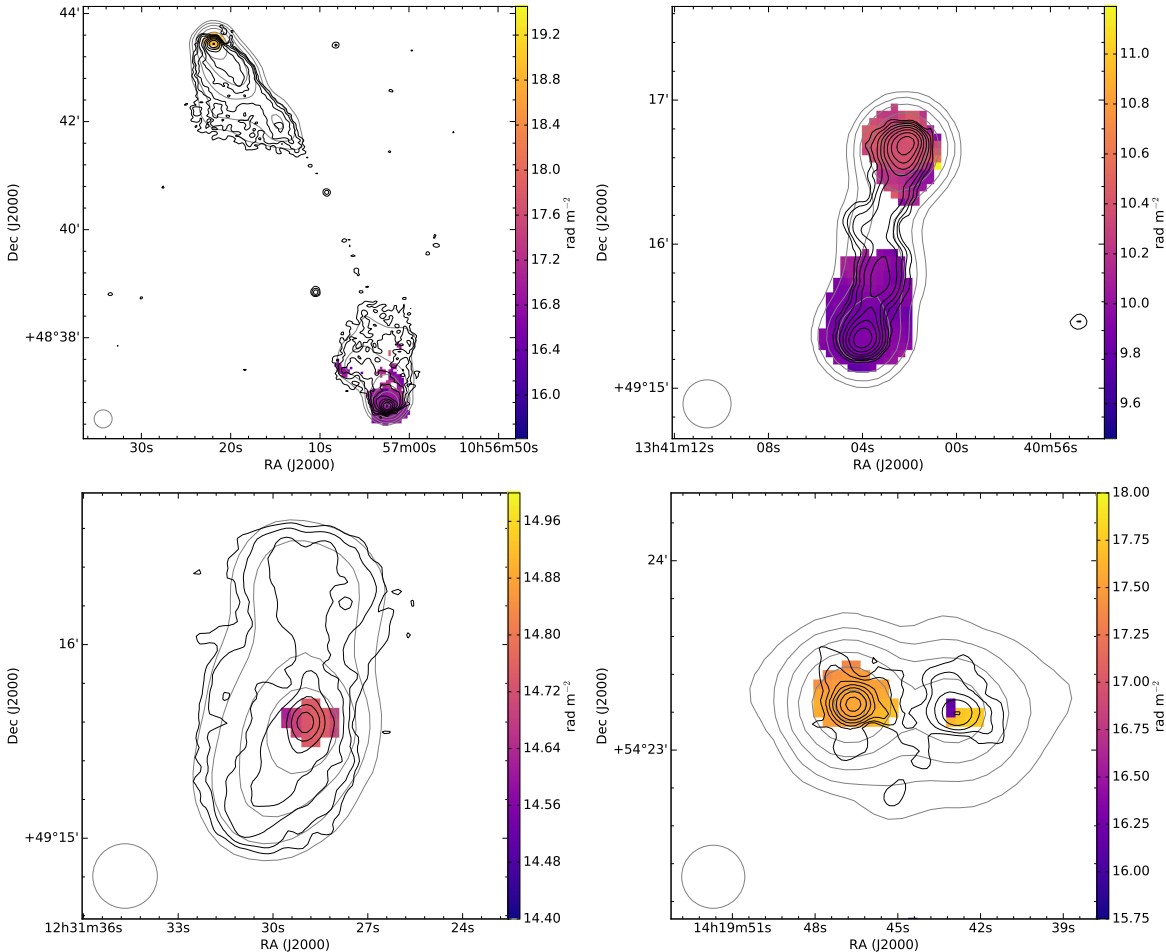

**Figure 4.** Total intensity and Faraday rotation images of four polarised sources in LoTSS-DR1. The grey contours and colour-scale Faraday RM images are at 20″ resolution (the convolving beam is indicated by the circle in the bottom left corner of the images), while the black contours are at 6″ resolution. Contours increase by factors of two, with the low contour for each source quoted below. The RM images are clipped at eight times the noise level in Stokes $Q$ and $U$. (**Top left**) FRII radio galaxy ILT J105715.33+484108.6, which has a projected linear size of ∼1.9 Mpc, and both hotspots are polarised. Low contour: 20″: 5 mJy beam$^{-1}$, 6″: 0.35 mJy beam$^{-1}$; (**top right**) FRII radio galaxy ILT J134102.90+491609.1, ∼1 Mpc in size, which is the most common type of polarised source found in the LoTSS-DR1 data. Low contour: 20″: 5 mJy beam$^{-1}$, 6″: 0.35 mJy beam$^{-1}$; (**bottom left**) only polarised source identified as an FRI radio galaxy in the current sample, with only the inner jet region detected in polarisation. Low contour: 20″: 5 mJy beam$^{-1}$, 6″: 0.35 mJy beam$^{-1}$; (**bottom right**) blazar ILTJ141945.53+542314.4, with both the core and the extension to the west being polarised. Low contour: 20″: 10 mJy beam$^{-1}$, 6″: 3.5 mJy beam$^{-1}$.

However, other types of radio-loud AGN were also present, for example, an FRI radio galaxy ($L_{144\,\mathrm{MHz}} \sim 8 \times 10^{24}$ W Hz$^{-1}$) with polarised emission detected only from the inner jet region (Figure 4, bottom left). Several of the compact polarised sources were also associated with blazars. Figure 4 (bottom right) shows a BL Lac object [46] with polarised emission in both the core and an extension of ∼200 kpc to the west. Indeed, the polarised emission in the kpc-scale extension exhibited some evidence for Faraday complexity with multiple peaks in the Faraday dispersion function at that location (Figure 6).

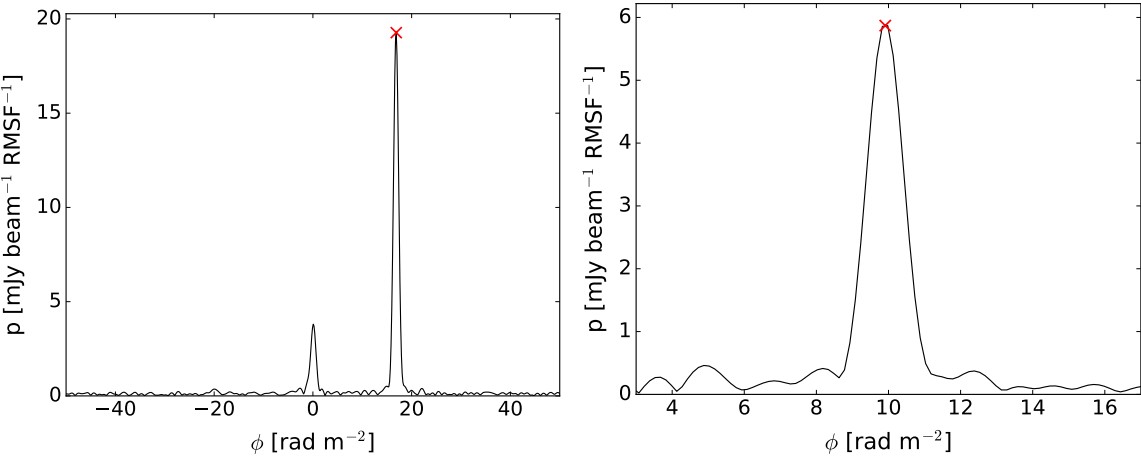

**Figure 5.** Examples of the absolute value of the FDF for the polarised hotspots of two FRII radio galaxies. (**Left panel**) FDF for the southwest hotspot of ILT J105715.33+484108.6 (Figure 4, top left), with the red cross indicating the real polarised peak of $\sim$19.3 mJy beam$^{-1}$ at a Faraday depth of $\sim$16.85 rad m$^{-2}$. The other peak in the FDF near 0 rad m$^{-2}$ is due to instrumental polarisation at a level of $\sim$2% of Stokes *I*; (**right panel**) FDF of the south hotspot of ILT J134102.90+491609.1 (Figure 4, top right), with a peak of $\sim$5.9 mJy beam$^{-1}$ at a Faraday depth of $\sim$9.9 rad m$^{-2}$.

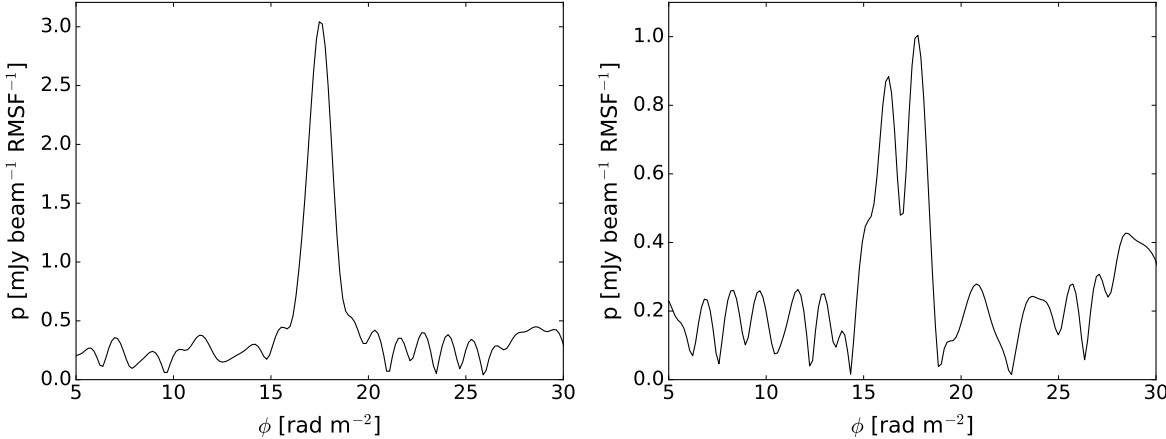

**Figure 6.** Example FDFs from blazar ILT J141945.53+542314.4 (shown in Figure 4, bottom right). (**Left panel**) Simple FDF of the blazar core with a polarised intensity of $\sim$3 mJy beam$^{-1}$ at a Faraday depth of 17.5 rad m$^{-2}$; (**right panel**) complex FDF of the western extension of the blazar, with a peak of $\sim$1.0 mJy beam$^{-1}$ at 17.7 rad m$^{-2}$ , and a second peak of $\sim$0.88 mJy beam$^{-1}$ at 16.2 rad m$^{-2}$.

## 4. Discussion

### 4.1. Nature of Extragalactic Polarised Sources at Low Frequencies

The most striking feature of the polarised sources in the LoTSS catalog is that the majority are large FRII radio galaxies, with a median linear size of 710 kpc. Indeed, polarisation observations at low frequencies could be a useful selection criterion for 'giant' radio galaxies, as $\sim$13% of all polarised sources have a linear size >1 Mpc. For FRIIs, the polarised emission is mainly associated with the hotspots. This means that the regions of detected polarisation extend well beyond their host-galaxy environment, and likely well into the outskirts of the galaxy group or cluster in which the host galaxy resides. As ionised-gas density and magnetic-field strength are known to decrease with radius in galaxy groups and clusters [47,48], this means that the effect of Faraday depolarisation is also much lower than near the centre [49]. As even small RM variations of $\sim$1 rad m$^{-2}$ across the emission region can be sufficient to depolarise a source below the detection limits at LOFAR frequencies, it is not

surprising that the LoTSS-polarised sources are so physically large. In addition, FRIIs are known to typically inhabit less-dense environments than FRIs [50–52].

A large linear size is likely not the only consideration here, as there are also large FRI radio galaxies [53]. However, in these sources brightness decreases with increasing distance from the inner jet out into the diffuse lobes [54] compared to FRII radio galaxies, where the brightest regions are near the outer edges of the source. Furthermore, even when imaged at 20″, the compact nature of FRII hotspots means that the emission probes a smaller Faraday depth volume than, for example, the extended lobes of an FRI radio galaxy. The large angular size of the polarised sources helps mitigate against wavelength-independent depolarisation, where variations of the intrinsic magnetic field and total intensity structure make the polarisation more difficult to detect in integrated measurements. Of course, it also helps to resolve more of the variations in Faraday rotation across the emission region. Figure 4 shows three such examples, where an additional polarised component is detected in the 20″ images compared to the 4.3′ images. More sensitive and higher-resolution observations may help detect the polarised emission from extended regions of FRI radio galaxies, unless internal Faraday rotation is strongly depolarising the emission in these sources.

### 4.2. Future Prospects

The ability to study the physical properties of extragalactic polarised sources and the intervening intergalactic medium is significantly enhanced by the unique capability of LOFAR to image at a high angular resolution (6″) at low frequencies, which is also crucial for the identification of the host galaxy. Soon, the WEAVE-LOFAR survey [55] will begin measuring redshifts for over a million of the LoTSS-detected radio sources. While only a small fraction of these sources will be polarised (possibly of an order of 10,000), it will still enable major advances in the field of cosmic magnetism, as it would become one of the largest RM catalogs with associated redshifts. Such large samples, coupled with the unrivalled Faraday depth accuracy, would enable the application of advanced statistical analyses to isolate the contributions of the different magnetoionic media along the line of sight to the net Faraday rotation and depolarisation [40,56–58].

**Author Contributions:** Data curation, S.P.O., C.L.V.E., M.J.H., M.H. and T.W.S.; Funding acquisition, M.B.; Investigation, S.P.O., M.J.H., T.W.S., C.T., V.V., C.H. and G.H.; Software, M.J.H., T.W.S., C.T. and G.H.; Writing—original draft, S.P.O.; Writing—review & editing, C.L.V.E., M.H., V.V., C.H. and M.B.

**Funding:** This research was funded by the Deutsche Forschungsgemeinschaft (DFG) under grant BR2026/23. LOFAR [16] is the Low Frequency Array designed and constructed by ASTRON. It has observing, data-processing, and data-storage facilities in several countries, which are owned by various parties (each with their own funding sources), and which are collectively operated by the International LOFAR Telescope (ILT) foundation under a joint scientific policy. The ILT resources have benefitted from the following recent major funding sources: CNRS-INSU, Observatoire de Paris and Université d'Orléans, France; BMBF, MIWF-NRW, MPG, Germany; Science Foundation Ireland (SFI), Department of Business, Enterprise and Innovation (DBEI), Ireland; NWO, The Netherlands; The Science and Technology Facilities Council, UK; Ministry of Science and Higher Education, Poland. Part of this work was carried out on the Dutch national e-infrastructure with the support of the SURF Cooperative through grant e-infra 160022 & 160152. The LOFAR software and dedicated reduction packages on https://github.com/apmechev/GRID_LRT were deployed on the e-infrastructure by the LOFAR e-infragroup, consisting of J. B. R. Oonk (ASTRON & Leiden Observatory), A. P. Mechev (Leiden Observatory), and T. Shimwell (ASTRON), with support from N. Danezi (SURFsara) and C. Schrijvers (SURFsara). This research has made use of data analysed using the University of Hertfordshire's high-performance computing facility (http://uhhpc.herts.ac.uk/) and the LOFAR-UK computing facility located at the University of Hertfordshire, supported by STFC [ST/P000096/1].

**Acknowledgments:** This paper is based (in part) on data obtained with the ILT under project codes LC2_038 and LC3_008. This research made use of Astropy, a community-developed core Python package for astronomy [59] hosted at http://www.astropy.org/; of Matplotlib [60], of APLpy [61], an open-source astronomical plotting package for Python hosted at http://aplpy.github.com/; and of TOPCAT, an interactive graphical viewer and editor for tabular data [62].

**Conflicts of Interest:** The authors declare no conflict of interest. The founding sponsors had no role in the design of the study; in the collection, analyses, or interpretation of data; in the writing of the manuscript; or in the decision to publish the results.

## Abbreviations

The following abbreviations are used in this manuscript:

| | |
|---|---|
| RM | Rotation measure |
| IGMF | Intergalactic magnetic field |
| GRM | Galactic rotation measure |
| FDF | Faraday dispersion function |
| FRI | Fanaroff Riley I |
| FRII | Fanaroff Riley II |
| AGN | Active Galactic Nuclei |
| BL Lac | BL Lacertae |
| LSS | Large-scale structure |
| LOFAR | Low-Frequency Array |
| HBA | High-Band Antenna |
| LBA | Low-Band Antenna |
| ILT | International LOFAR Telescope |
| LoTSS | LOFAR Two-Metre Sky Survey |
| DR1 | Data Release 1 |
| HETDEX | Hobby-Eberly Telescope Dark Energy Experiment |
| WEAVE | William Herschel Telescope Enhanced Area Velocity Explorer |
| APERTIF | Aperture Tile in Focus |
| ASKAP | Australian Square Kilometre Array Pathfinder |
| EMU | Evolutionary Map of the Universe |
| POSSUM | Polarisation Survey of the Universe's Magnetism |
| NRAO | National Radio Astronomy Observatory |
| VLA | Very Large Array |
| NVSS | NRAO VLA Sky Survey |
| VLASS | VLA Sky Survey |
| SDSS | Sloan Digital Sky Survey |

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
