# Peer review of "Untangling Cosmic Magnetic Fields: Faraday Tomography at Metre Wavelengths with LOFAR"

_galaxies, doi:10.3390/galaxies6040126_

Reviewer 1 Report

I enjoyed reading this manuscript and have only a few suggestions for minor edits which are contained in the attached file 'revisions.pdf'. After these small alterations have been made I'd be happy to approve this manuscript for publication.

Author Response

### We thank the referee for their careful reading of the paper. All requested changes have been addressed below.

Reverse the order here to match the first part of the sentence: 'from 110 to 250 MHz with High Band Antennas (HBA).’

### Done

'RM' -> 'rotation measure (RM) . I think this is the first instance of 'RM' in your paper - you should define the acronym here.

### First occurrence is on line 28. In any case, changed to “…time-dependent ionospheric Faraday rotation correction…"

'that' -> than'

### Done

dashed 'blue line

### Done

Consider bold facing this point for emphasis.

### Didn’t use bold font, but repeated this in Section 4.1 for emphasis: “Indeed, polarisation observations at low frequencies could be a useful selection criteria for 'giant' radio galaxies, as $\sim$13\% of all polarised sources have a linear size $>1$~Mpc.” and in the abstract: “ (with $\sim$13\% of all sources having a linear size $>1$~Mpc)"

dashed 'blue line

###  Done

Do I follow you correctly then that if the observed gradient in rotation measure between the lobes is entirely due to intervening IGMF there is a tension between the estimated magnetic field strength of the intervening plasma (~micro Gauss) and the sorts of field strengths present in simulations of the IGMF within LSS filaments (~nano Guass).

### Yes, more like a tension between 100’s of nG (obs) and 10’s of nG (in simulations). But better GRM reconstructions are needed for smaller scales, which will probably explain this tension. 

Modified text: “The magnetic field in the simulation was amplified to between 10 and 50~nG from a initial magnetic field of 1~nG..."

a relatively low

### Done

Perhaps consider re-structuring these sentences a little:

"Despite this limitation, however, smaller datasets have been made available for a select number of individual sources which can then be imaged at higher angular resolution (~20"). In Figure 3, we present examples of several sources for which polarized emission was detected within these higher angular resolution images.”

### Reworded to: “To circumvent this limitation, smaller datasets have been made for the polarised sources (by phase-shifting the uv-data to the source coordinates and averaging in time), which were then imaged at the higher angular resolution of 20", following the procedures described in \cite[e.g.]{neld2018,osullivan2018}. “

The grey and black contours are hard to distinguish visually. Could you perhaps use another color (say green) for the 20" resolution contours?

### No, this was kind of the point. To avoid visually cluttering the image. The black contours better show the source structure so this is what I want the reader to see clearly, but the grey contours are there mainly just to explain the extent of the RM pixels. 

After the discussion in Section 3.3 I thought you would want to show us the image of ILTJ123459.82+531851.0?

### Yes, now I have done so. 

the convolving beam is indicated …

### Done

as FRIIs

### Done

with polarized emission in …

### Done

the polarized emission in the ...

### Done

ILT J141945.53+542314.4 (shown in Figure 3, bottom right)

### Done

sources 'in the LoTSS catalogue'

### Done

and will become

### Done 

Reviewer 2 Report

The paper summarizes the results of Faraday rotations which obtained from LOFAR low frequency observations,  which are new and important for understanding of the polarizations of galaxies and AGN at low frequencies. Probably due to page limit, some details, e.g. the resulted or estimated magnetic fields along the sources were not introduced, and also the double peaks in the blazar's FDF were not explained or discussed in some way in the paper.

Line 108: ..are shown..

Author Response

The paper summarizes the results of Faraday rotations which obtained from LOFAR low frequency observations,  which are new and important for understanding of the polarizations of galaxies and AGN at low frequencies. Probably due to page limit, some details, e.g. the resulted or estimated magnetic fields along the sources were not introduced, and also the double peaks in the blazar's FDF were not explained or discussed in some way in the paper.

### We thank the referee for their careful reading of the paper and the comments. We have not expanded too much into the details in some cases as the goal for this paper was to give an overview of what has recently been done in this area, and to give some indications of future topics that are worth exploring further. 

Line 108: ..are shown..

### Corrected.